# The Meaning of Physical Literacy for Instructors of Children Experiencing Disability, from an Ecological Systems Perspective

**DOI:** 10.3390/children10071185

**Published:** 2023-07-07

**Authors:** Kyle Pushkarenko, Janice Causgrove Dunn, Donna Goodwin

**Affiliations:** 1School of Human Kinetics and Recreation, Memorial University of Newfoundland, St. John’s, NL A1C 5S7, Canada; 2Faculty of Kinesiology, Sport and Recreation, University of Alberta, Edmonton, AB T6G 2R3, Canada; jcausgro@ualberta.ca (J.C.D.); dgoodwin@ualberta.ca (D.G.)

**Keywords:** children, disability, ableism, interactive influences, interpretive phenomenological analysis, community-based programming

## Abstract

With the rapid and widespread uptake of physical literacy (PL), there is potential for instructors to devalue participation of children who experience disability. The aim of the investigation was to understand how instructors who facilitate physical activity for children experiencing disability make sense of PL, and more specifically, how these instructors bring meaning to PL. Using interpretive phenomenological analysis, six instructors engaged in individual, semi-structured interviews. The study rationale was underpinned by the conceptual framework of ecological systems theory, which provided a foundation for the research, guided the structure of the interview guide, and facilitated a reflexive interpretation of the findings. Four themes were generated: Recognizing unique embodiments, The importance of context, Beyond physical competence, and Navigating PL’s dominant discourse. The instructors’ meaning of PL, impacted by relational and environmental influences, reflected the importance of movement skill development, while also embracing diverse embodiment and pedagogical flexibility by giving value to exploratory play, partial participation, family involvement, and willingness to abandon rigid lessons plans.

## 1. The Meaning of Physical Literacy for Instructors of Children Experiencing Disability, from an Ecological Systems Perspective

Since its inception in the late 1800s [1], multiple conceptualizations of physical literacy (PL) have emerged. Consequently, understandings of PL and its application to movement settings have varied greatly [2]. Most understandings of, and approaches to, PL share commonalities grounded in a Whiteheadian perspective [3,4]. 

Whitehead defined PL as “the motivation, confidence, physical competence, knowledge and understanding to value and take responsibility for engagement in physical activities for life” ([5], p. 8). A concept, comprising interacting developmental domains (i.e., affective, cognitive, physical, and behavioural), PL contributes to the development of the whole person, both mind and body as one [6]. Further, PL is defined as “the set of capabilities and dispositions that enable active participation in movement cultures across an individual’s lifespan” ([7], p. 1). through unique and embodied interactions with the outside world [4,8]. PL is deemed essential to all individuals’ overall development and quality of life [9,10]. 

With the declaration that PL is inclusive of all [5,6], organizational leaders affording services to children experiencing disability (We use the term experiencing disability to “acknowledge the wide variety of embodied sensations, social structures, cultural understandings, and identities that may be related to someone’s disability experience” ([11], p. 275)) have adopted PL as foundational to programs aimed at fostering a lifelong love for, and engagement in, physical activity [12,13]. Unfortunately, the popularized understandings (i.e., dominant narrative) of PL being embraced and incorporated into these programs may not address the diverse embodiment requirements of children experiencing disability [14]. Rather than emphasizing something that is unique, relational, and holistic, conceptions of PL emphasized through education and training, professional development opportunities and mandatory organizational training reflect a one-size-fits-all approach to the concept [15,16], privileging physicality in the form of acquiring set of motoric skills considered to promote excellence and ongoing participation in physical activity [17,18]. As such, programs designed to meet the specific needs of individuals experiencing disability tend to, either consciously or unconsciously, overlook the holistic Whiteheadian perspective that PL rests on and instead apply normatively based movement skill standards (i.e., competency standards) to their instructional goals [19]. This “body-as-machine” ([15], p. 107) operationalization represents an uncoupling of PL from its core meaning fostering development from the aspect of physical competence [20], increasing the potential of fostering physical activity climates that lead to the devaluation of those with diverse embodiments, reinforcement of ableist ideals (Ableism is “… system of causal relations about the order of life that produces processes and systems of entitlement and exclusion. This causality fosters conditions of microaggression, internalized ableism and, in their jostling, notions of (un) encumbrance. A system of dividing practices, ableism institutes the reification and classification of populations. Ableist systems involve the differentiation, ranking, negation, notification and prioritization of sentient life” ([21], p. 288)), and narrowing of opportunities for meaningful physical activity engagement [16,18]. A prime example of this is the interpretation of PL utilized in the Long-Term Athlete Development Model (LTAD) developed by Canadian Sport for Life [22], whereby PL is narrowed to bear “only the faintest resemblance to the root definition” ([18], p. 955).

As individuals experiencing disability possess a wide range of personal interests and capabilities [23], we can assume that the understanding and meaning instructors give to PL in this context may necessarily vary from that of those who do not experience disability. In other words, it is not reasonable to discuss and promote a concept emphasizing individual expression based upon one’s unique embodied potential while not taking into consideration different life experiences and the meaning of PL by instructors of children experiencing disability [5,23]. We have yet to reflexively explore the ambiguity in our understandings of PL for children experiencing disability [14]. Doing so is essential for fostering the operationalization of PL based on Whitehead’s [5,6] perspective and the idea that the PL journey occurs according to each person’s embodied potential [8].

Yi et al. [24] suggested that context-specific understandings of PL from various key stakeholders are essential to ensuring meaningful physical activity experiences are had by all. At the intersection of physical activity and disability, adapted physical activity (APA) instructors can contribute valuable insight into context-specific understandings of PL [25]. APA instructors not only possess the knowledge to create “disability-friendly environments” ([25], p. 449) through their use of teaching, communication, and adaptation strategies, they also promote choice and autonomy, person-centred, and ethically reflexive opportunities for meaningful participation [26,27]. APA instructor’s perspectives are imperative to understanding the full meaning of PL, what it is, and what it comprises [17,28]. The aim of the research was to ascertain how instructors make sense of PL. Specifically, the research objective was to understand how instructors who facilitate physical activity for children experiencing disability bring meaning to PL. 

## 2. Conceptual Framework–Ecological Systems Theory

Jurbala [10] proposed that PL requires an understanding of the ongoing and reciprocal interactions between each of its developmental domains (i.e., affective, cognitive, physical, and behavioural) [5,6] in accordance with one’s experiences with both physical (i.e., the built environment) and social (i.e., parents, peers, instructors) influences [29,30]. To highlight the various layers of social and environmental influences on the meaning of PL, Bronfenbrenner’s [31,32] ecological systems theory (EST) served to provide a reflexive framework for the generation of the research question, the creation of the interview guide, and as a context for data analysis and interpretation.

EST is a view of human development based upon the dynamic interplay between the organism and the environment, organized into five environmental levels: microsystem, mesosystem, exosystem, macrosystem, and [31,32]. Each system contains a set of influences that may positively or negatively affect development. Most proximal to the individual, the microsystem consists of the relationships between individuals and their immediate environment (e.g., parents, siblings, peers, teachers, and instructors). As a result of this proximity, it has the most immediate and impactful influence [33]. The mesosystem contains the interactions between an individual’s microsystems (e.g., the interactions between parent and practitioner, or the surrounding physical infrastructure). The exosystem symbolizes a context in which the individual is not actively participating [34,35]. Here, influence extends from a social structure within contexts with which one engages (e.g., workplace culture, program staff attitudes and conduct). The macrosystem refers to the institutional patterns existing within an individual’s societal culture [31]. It is the overarching collection of characteristics embedded in the previous three systems, including societal belief systems and values, political and economic structures, cultural customs and principles, and lifestyle patterns [31,35,36]. In other words, a particular culture, subculture, or context’s societal blueprint [36]. Finally, the chronosystem consists of the life experiences occurring over the lifespan, inclusive of significant life transitions (e.g., physical development and maturation), organismic events (e.g., illness, injury), and sociohistorical happenings (e.g., pandemic health restrictions) [35].

## 3. Method and Methodology

With the aim of the research being subjective understanding of how instructors make sense of PL, an interpretivist paradigm, founded on the assumptions that reality is socially constructed and fluid, was required [37]. An interpretivist paradigm is based on a relativist ontology, transactional/subjective epistemology, and hermeneutic and dialectical methodology [37,38]. We assume that multiple experiential realities coexist, acknowledging the subjectivity of knowledge and the co-creative nature of understanding. This recognition arises from the interactive and interpretive process that occurs between the researcher(s) and the subjective experiences of the participants. Furthermore, we acknowledge that our actions and decisions embedded in our positionality inevitably impacted the research processes and outcomes [39,40]. We acknowledge that we are experienced qualitative researchers, White, and nondisabled, with backgrounds in professional practice and postsecondary teaching of APA. The first author was familiar with the program where the research occurred and four of the six instructors who ultimately participated in the study. 

Consistent with an interpretivist research paradigm and its assumptions, an interpretative phenomenological analysis (IPA) research approach was applied [41]. IPA provided a systematic means to explore and understand the subjective, lived experience of the participants as they made sense of their personal and social worlds [42,43]. IPA has three unique, interwoven influences: phenomenology, hermeneutics, and idiography [41]. Phenomenology consists of how individuals derive meaning from their experiences (or phenomenon), making them theirs alone, exclusive, or distinctive from others [41]. Hermeneutics is the study of interpretation and meaning [42]. In essence, researchers aim to understand how participants value and interpret their experiences. Finally, idiography includes the analysis and development of detailed accounts of each participant or case [41]. Within IPA research, each participant undergoes extensive analysis to ensure a sense of value for their diverse or variable experiences [43,44] before moving on to patterns of similarities and discrepancies from the collective group [41].

After obtaining approval from the University’s Institutional Ethics Review Board, recruitment letters were sent to community PL leaders (i.e., organizational representatives) outlining the specifics of the research investigation. Organizational leaders then distributed the recruitment letters to their organization’s APA instructors, who if interested, contacted the first author to become involved in the study. The first author invited those who met the eligibility requirements to participate in the research. Before data collection, participants were provided with study information, inclusive of the benefits, risks, time commitment, and study withdrawal procedures. All participants provided written informed consent prior to their involvement. To safeguard anonymity, all identifying information was carefully eliminated from the research materials, and pseudonyms were assigned to each participant.

### 3.1. Participants

In accordance with IPA, a small, homogeneous, purposeful, criterion-referenced sample provided in-depth exploration of the phenomenon [41,45,46]. The six instructors recruited provided data adequacy and in-depth idiopathic analysis of their experiences [47]. The inclusion criteria were (a) provided community-based APA instruction to children experiencing disability between the ages of 7 and 12 years over the past 24 months; (b) held the role of instructor for six consecutive months or greater within the organization over the past 24 months; and (c) were willing to speak about their experiences of PL in the instruction of children experiencing disability. The age range of 7 to 12 years was selected because children experiencing disability are the most susceptible to activity decline in this period [48]. The duration of 24 months offered a hermeneutic experiential timeframe, allowing participants to recollect and contemplate their instructional experiences. The criterion of six consecutive months as an instructor was chosen as the minimum amount of time needed to become accustomed to a program and its participants, enabling the formation of a point of view about the phenomenon of PL. For the final criterion, it was assumed that instructors who taught within a program framed by PL development would be able to articulate perspectives on the concept of PL.

Despite efforts to recruit APA instructors from multiple programs within a large Canadian city, all participants (two male and four female) were from a single program at a single location. Their average age was 27 years (range: 22 to 35 years), while the average number of years instructing in the organization was 3.9 years (range: 0.5 to 6.5 years). Four participants had previous professional experience working with children experiencing disability as a respite care worker, educational assistant, or Special Olympics coach. All participants had a background in APA having taken one or more undergraduate classes in APA.

### 3.2. Community Program

The participants were instructors at a not-for-profit, disability-specific, community-based physical activity and fitness centre in a large Canadian city. Several programs are offered at this centre, including one designed for children aged 4 to 19 years with the designated label of developmental disabilities. Within this program, the children received one-on-one and group instruction 30 min twice per week over a period of 8 weeks. Focusing on the development of movement skills and knowledge towards health and wellness while facilitating autonomy and independence in a fun and social environment (i.e., play-based context), the program is facilitated by instructors educated and trained on PL development according to the various stages of the LTAD [22]. A typical instructional session included a warm-up game, two to three individualized activities emphasizing fundamental skill development and directed towards each participant’s unique goals and objectives, and a cool-down period (i.e., free play). Parents attended the programs and were present in case support of their children was required. However, their primary role was that of an observer. 

### 3.3. Data Collection

The participants engaged in one-on-one face-to-face, audio-recorded, semi-structured interviews. All interviews were conducted by the first author at a time and place convenient to all of the participants. The interviews followed an interview guide that was developed based on the research focus and conceptual framework, thereby ensuring consistency of inquiry across participants [46]. As recommended by Smith et al. [41], the interview guide contained descriptive (e.g., What does PL mean to you?), narrative (e.g., How did you arrive at your understanding of PL? From your experiences, what sorts of activities do you feel are most beneficial for developing a child’s physical literacy?), and structural questions (e.g., How do things like knowledge and understanding, or motivation and confidence fit in with your understanding of physical literacy?). To ensure the conceptual and methodological framework was represented and that questions provided data relevant to the research objectives, two experts in qualitative research provided written feedback on the ordering of questions, relevance of questions to the research objective, depth of questions, and clarity of the interview guide [49]. 

Initial interviews with participants lasted an average of 59 min (50 to 75 min). Follow-up interviews, used to clarify information from the initial discussions, occurred with four of the six participants and lasted an average of 20 min (10 to 45 min). After each interview, the first author recorded his thoughts in the form of a reflexive journal. This action of journaling included thoughts about the conversation, the interviewer’s reflections on the participants’ discussions, and considerations for potential emerging themes. The journal notes enabled a reflexive return to the interview setting during analysis [50,51]. The process further facilitated scrutinization of researchers’ positionality [52] and “factors influencing the researcher’s construction of knowledge” ([53], p. 275). 

### 3.4. Data Analysis

The first author completed an initial inductive, line-by-line thematic analysis of the interview transcripts using Smith et al.’s [41] six-step framework. The process of analysis began by immersing oneself completely in the data of the first participant. This included listening to the audio recordings of the interview, reading and rereading the corresponding transcript, and reviewing the reflexive journal entries. Secondly, descriptive, linguistic, and conceptual notes regarding verbal and nonverbal behaviour [e.g., idiographic word choices, tone of voice, and body language; 41] were noted in the margins of the transcript, subsequently typed up, and transitioned to a separate document. Third, once completed, the typed notes were, printed out, cut up, and physically arranged to identify initial themes. Fourth, connections across the themes occurred. Fifth, when this process was completed, it was repeated with the data for each of the remaining participants. In the sixth and final step of the analysis, patterns across participants were explored to identify the key themes of the collective group, deepening the analysis by utilizing the conceptual framework as a lens to interpret the findings. 

The second and third authors acted as “critical companions” ([54], p. 340) for reflexive discussions of data analysis and theme generation, interpretation of, and presentation of the findings [54,55]. Discussions continued until there was shared understanding among the authors. Drawing upon Bronfenbrenner’s [31,32] EST, the authors were then able to finalize a profound comprehension of the individual meanings held by each participant, and the shared meanings within the collective group [56]. 

In addition to the participants receiving via email, a copy of their transcript(s) for review [57], a summary of each participant’s preliminary themes was emailed to them with an invitation to explore gaps, share insight and comment on the interpretations generated by the research team [i.e., member reflexions; 55]. All six participants accepted the invitation. With the exception for one person who added clarifying comments to their transcripts, no requests for changes occurred. 

### 3.5. Trustworthiness

Four criteria were used to convey the quality of the research thereby fulfilling a commitment to “integrity, competence, and legitimacy of the research process and findings” ([58], p. 194). The criteria were (1) sensitivity to context, (2) commitment and rigor, (3) transparency and coherence, and (4) impact and importance [41,59]. The authors achieved sensitivity to context by remaining aware of the theoretical, philosophical, and methodological underpinnings of the research process throughout the study’s entirety [59]. This process included understanding the relevant literature and previous application of EST to physical activity settings and acknowledging researcher and participant power structures by declaring the researchers’ positionality. The authors actively engaged in reflexive discussions during the formulation and finalization of themes, interpretation of the findings against the conceptual framework, and the discussion of the meaning of the findings and implications for professional practice [40]. Moreover, sensitivity occurred through purposeful sampling thus adhering to an idiographic commitment made to the participants’ experiences [41,58]. 

Commitment and rigour were established through use of multiple data sources (interviews, reflexive journal), comprehensive data collection (multiple interview opportunities), and systematic analysis strategies (six-step process). Further, the participants were provided with opportunities to reflect on their interview transcripts and thematized data, adding a supplemental degree of credibility [58]. Transparency and coherence were upheld by offering a comprehensive description of the research methodology and ensuring complete disclosure of the participants’ roles prior to their involvement in the study [58,59]. Reflexive journaling was used to acknowledge personal influences on the knowledge generated and bring depth of understanding to the findings [40] and its interpretation through the lens of EST [41,59]. Pre-existing knowledge of the program and instructors assisted in developing rapport with the participants; however, this relationship was documented, and reflexively discussed with the participants and amongst the researchers. Participants were reminded that their interview information would not be shared with their supervisor (beyond quotations that may appear in a potential publication) [58,59]. 

Impact and importance are contingent upon the reader’s subjective evaluation and the subsequent actions they take based on the study findings. By providing a meticulous account of the research processes, research context, and richly descriptive participant experiences supported by direct quotations, readers can assess the relevance of the findings to their own context, and make informed decisions about how to apply them [57,60].

## 4. Findings

Four themes generated from the analysis reflected relational and environmental influences on the meaning of PL held by the instructors. These influences occurred at the intersection of individuals (microsystem) with their environment (mesosystem), the cultural (sport) context of PL (exosystem), and the larger interaction of social and cultural values embedded in (normatively referenced) active lifestyle development (macrosystem). The instructors understood PL through an ongoing negotiation between best pedagogical practices emphasized within the field of APA and the playground to podium sport foundation of PL (CS4L, 2016). The themes were: (a) Recognizing unique embodiments, (b) The importance of context, (c) Beyond physical competence, and (d) Navigating PL’s dominant narrative.

### 4.1. Recognizing Unique Embodiments

Although adhering to a movement competency understanding of PL, the instructors collectively understood that each child expressed unique embodiments, abilities, interests, and needs and that their (microsystem related) relationships with the children were immediate and impactful. Rachel highlighted the need to step away from the macrosystem influences of PL (interaction of micro-, meso-, and exosystems of macrosystems of influence), that being norm referenced movement development with the aim of sport focused goals, to encourage participation of children with diverse embodiment. “I think that physical competency looks so different for everyone”. Patrick also noted that competency depended on one’s embodied capability: “It [motor ability] is unique to everyone, 100%. Everyone is going to have a different ability level when it comes to a certain movement or skill when using their body in time and space”. Beth further added: 

To one person, it [PL] might mean that they can complete a triathlon. To another person, it might be that they can get by in gym class and not feel like they are being pointed out or singled out for not being good at that particular game or something.

Instructors highlighted important aspects of adaptations they make that may not be present in non-APA or inclusive settings (e.g., deviating from planned lessons, taking participant interests into account, valuing partial participation, and meeting the children where they are emotionally and physically on a particular day). Samantha expressed the need to adjust for differences in movement ability and overall physical proficiency; however, acknowledged that there is “no definitive recipe that can be followed”. Beth underscored the need to deviate from planned activities to prevent or alleviate expectation frustrations that can impact children’s motivation to continue engaging with planned activities. “… if I see a kid getting really frustrated with a skill or something, I’ll just say, ‘you know what, let’s just play a game, and then we’ll go back to it’”. Incorporating interests were deemed important by the instructors to enhance children’s enjoyment, ultimately increasing the meaningfulness attached to a PL experiences. Beth recalled, 

It is one thing to just hit a ball and then run around the bases, but that’s pretty boring for a kid who’s not really interested in sports. But if you make it exciting for them by bringing in extra visuals and noises and things that they enjoy, just making any game more enjoyable and more individualistic, that’s going to make them want to come back.

Further, they valued partial participation above overwhelming a child by forcing engagement in a non-preferred activity. Andrea emphasized meeting the children where they were at by honouring what they would be willing to do rather than imposing what instructors may have felt they needed to do (i.e., encouraging some form of positive engagement in physical activity over no engagement at all). 

I had a kid who was really dysregulated all day, and I was getting to the point where I thought, ‘I’ve tried all of my measures,’ but we just sat there and passed the ball back and forth. He did nothing of what I planned to do, but at least he was still doing some type of physical, you know, activity. 

Martin noted that flexibility in the presentation of information was important, highlighting that for some children, simplifying activities was necessary to help them remain engaged and “not being overwhelmed with information”. Instructors also discussed the need to know how participants felt physically as they entered the physical activity environment. Patrick highlighted how a child’s physiological state when entering the physical activity environment is a pedagogical strategy that gives value to partial participation and establishes a positive, participatory context. “There are days when kids come in exhausted, and you know what? You just slow things down. You do what you can”.

### 4.2. The Importance of Context

Instructors emphasized the importance of constructing environments with the participant in mind thereby creating positive PL contexts (mesosystem influences of children, instructor, and environment). Beth and Samantha highlighted the need to address certain physical characteristics of the environment such as noise, people, and colour. According to Beth, attending to these features, helps to establish a level of comfort by “reducing the risk of over-stimulating the individual”. In addition to accommodating noise, Andrea offered up suggestions for making the physical environment easier to navigate, highlighting her use of “headphones for the purpose of noise reduction, and the use of visual markers to assist with transitions from one activity to the next”. Finally, Patrick indicated that contexts created according to individualized needs provided “a sense of overall security”, resulting in a learning climate that is welcoming to children. This comfort contributed to an “optimal experience, one that individuals positively associate with, and want to come back to”, thus maximizing the opportunity for further PL engagement. 

Beyond discussions of the physical environment, instructors also believed that their attitudes profoundly impacted the meaning they gave to PL. Samantha indicated that instructor attitudes of “acceptance of diversity and inclusion, allowing children the opportunity to come into the physical activity setting without judgment”, fostered the best possible learning climate for children, strengthening their overall motivation to engage. 

Providing opportunities without a rigid movement-skills focus represented a welcoming and supportive physical activity context. As Rachel expressed, physical activity opportunities should occur within a supportive, enjoyable, success-oriented context. Although the instructor may have a PL goal in mind, “if kids feel like it is work, it’s going to be viewed as negative”. She believed that opportunities should allow children to have the freedom to play and have fun without worrying about completing an instructor imposed developmental movement task or accomplishing an instructor driven goal. Reflecting on her experiences, Samantha indicated that using play as a pedagogical strategy allows one to “experiment and try things out without someone telling them what’s right or wrong”, from which further instructional activities and strategies can emerge. Samantha highlighted that eliminating pressures to meet criterion-framed normative movement skill standards nurtured a context of happiness, fun, and success in movement. As Andrea put it, play increases the child’s confidence by developing an “understanding of what it means to be physically active”. 

### 4.3. Beyond Physical Competence 

There were dynamic interactions among the influences instructors gave to their meanings of PL for children experiencing disability. The instructors gave importance to relationships with the children and the parents in the immediate environment (microsystem influences) as well as the community at large (macrosystem influences). The instructors held a steadfast belief that others’ perspectives were essential in optimizing their understanding of PL as it was applied to children experiencing disability. Parents were a vital component of the program context, providing information on their children’s strengths, interests, and motivations and were considered to be part of the instructional team-approach (interactive mesosystem influence). Thus, a team-based approach consisting of regular parental consultation elevated the quality of PL experiences for the children. An ongoing dialogue allowed instructors and parents to exchange information. In describing her interactions with parents, Beth indicated that parents brought meaning to her understanding of PL by establishing consistency across different environments in the children’s days. 

A few parents have suggestions like “maybe you can try this; it has worked in the past” and we say, “great!” We will try that, and like visual schedules, sometimes parents really recommend we do that because they rely on one at home. We say, “okay, we will for sure use a visual schedule then”.

Similarly, Rachel shared how conversations with parents enhanced the meaning of PL by enhancing important and productive relationships with the child. She embraced the microsystem influences of parent input on learning preferences and ultimately the meaning of PL for specific children. 

I talked to mom, and she said often her little boy needed (deep neurological) pressure when he was overwhelmed or upset. If we were doing a task that might be more challenging for him, he would just come and sit on my lap which meant he needed a squeeze.

In addition to seeking contributions from parents, the instructors shared the importance of children’s contributions. They valued the importance of activity preferences, providing choice in equipment selection, pacing of the activities, and asking for feedback on their experiences. Patrick noted, however, that obtaining the children’s input was challenging, “… it is sometimes very difficult to get feedback from them because of the communication limitations and cognitive abilities”. Beth explained that providing children with choices was an optimal way to work around variations in communicative ability and afforded the children with a sense of direction in creating future opportunities that may increase relevance and importance of active movement participation. Beth recalled:

… the parents really wanted him to work on body awareness, and they thought yoga was a great way to do it, but he hated it. So, we just started playing a different game and rolling dice, and whatever number appeared, he had to come up with a movement to do that many times … it was just finding different ways to do the skill.

Finally, instructors highlighted that PL is not isolated to a single context. Input and support from those within the community were necessary to reinforce the overall and holistic meaning of PL (macrosystem levels of influence). Rachel indicated that discussions with community members within various sport, recreation, and physical activity sectors were “interesting” as they opened her eyes to new considerations regarding her instructional approaches. Specifically, these conversations stimulated opportunities on how her current approaches could accommodate a smoother, more continuous transition to other physical activity settings that integrated their own cultural customs and value systems to participation. After all, according to Rachel, “it all comes down to exposure and development, or giving meaningful opportunities to the kids, so they can be active in the future”. Patrick went one step further, articulating that, not just conversations, but the relationships formed by instructors across contexts was essential for fostering consistency and continuous PL development:

As far as fostering physical literacy, kids need to be in an appropriate environment, inclusive of those that interact with them. Building strong and long-lasting relationships with other facilitators, those that are willing to share information, will provide better results in developing physical literacy from one place, or program, to the next.

Addressing continuity from a slightly different perspective, Samantha articulated that community support was essential for the continuous development of PL. With it, long-term engagement in physical activity is possible. Expressing subtle bitterness, she conveyed that the current system of services for children experiencing disability is inadequate as there is a lack of options once they reach a certain age: “… it’s really difficult once they leave our program. Once they age out, if we don’t have something set up in the community or there are no community-based programs, there is nowhere they can go”. This discontent resonated through Martin’s observation that the amount of time individuals spend with him is limited. For PL development to evolve, “it is something that needs to be constantly and consistently reinforced by individuals beyond the physical activity context”. 

### 4.4. Navigating PL’s Dominant Narrative

The instructors reflected the influence of the cultural value (macrosystem level) of normatively based, movement competence as the base of the sport development framework of PL (CSFL, 2016). Participants highlighted that the acquisition of movement skills were foundational for the improvement of overall movement ability and hence PL. For example, Rachel declared that movement skills served as “the basis for informing physical literacy”, suggesting they are imperative for ongoing participation in physical activity. Beth stated that movement skills provided an increased opportunity for children experiencing disability to “know how to move or play almost any game or sport, in any environment”. She further emphasized that by encouraging movement skill development, children were better prepared to participate in physical activity:

We want them to be able to more easily know how to run, walk, skip, oh my gosh, all the different physical literacy tools and like especially the fundamental movement skills so that if they, you know, work on stair climbing or running or jumping in our centre then that translates to like the home and the gym.

Not unlike Rachel and Beth, Patrick indicated that foundational movement skills are the basis for all types of future engagement in physical activity:

Physical literacy is like learning how to read and write where you first learn the alphabet—the A, B and Cs. With physical literacy, your ABCs are learning basic movement skills like running, jumping, hopping and skipping … these skills support a range of physical movements from recreational capabilities to becoming an athlete.

Martin further noted that movement skill acquisition fostered a sense of independence and self-sufficiency, improving one’s “ability to self-guide their physical activity engagement, leading to participation in more advanced activities”. 

The instructors further reaffirmed the dominant narrative associated with PL through the considerable value they placed on movement skill assessment—the social value placed on categorizing competence (exosystem influences imposed on the children experiencing disability through instructor attitude and action). Patrick’s explanation summarizes a shared understanding among the participants that assessment was valuable for providing an initial indication of the movement competence of children experiencing disability. However, rather than seeking to document deficits based on norm-referenced performance standards, assessments were used as a tool for self-referenced instruction. The assessments provided a frame of reference to begin instruction and provided “a measure of progress, or more specifically, a measure of change”. For the participants, movement skill assessments were essential for generating a long-term, customized instruction plan, complete with information on optimizing PL development. 

## 5. Discussion

The application of EST [31,32] revealed that instructors’ understanding of PL was shaped through multiple layers of influence occurring as a result of their ongoing relationships and interactions with the physical and social world [10]. Beyond the conceptual knowledge received as a result of their mandated organizational education and training, the meaning that instructors attached to PL yielded an understanding that extends far beyond the likes of physical competency and the dominant PL narrative organizations commonly adopt into programming, and thus, often persists through practice. 

Despite noting the importance of the ‘physical’ (i.e., fundamental movement skill development) as being foundational to PL development [17], proximal influences (e.g., microsystem influences of children and parents) articulated by instructors, were reflective of understandings that were increasingly complex. Much to their credit, instructors understood PL as being something uniquely expressed according to an individual’s embodied capability [5,6]. They recognized that holding rigid to a normative understanding of physical competence held greater potential to reinforce the ableism and exclusion so often encountered by people experiencing disability in physical activity settings [16,61]. 

Similarly, instructors acknowledged the variability that occurs with learning. Noting the need for person-centred flexibility and variation in instructional approach (e.g., partial participation, choice, play-based instruction), and that participants should be empowered to guide instruction rather than being a pure recipient of it, instructors highlighted a necessary departure from linear, readiness-based styles of PL development (e.g., exosystem and macrosystem influences of sport and broader society; [15]). In doing so, instructors avoided reproducing an ableist narrative, ‘othering’ those deemed to be ‘unable’ [16], remained true to the underpinning that PL is an inclusive concept [5,6], and expanded upon current denotated conceptualizations of PL by bringing relevance to the notion of ‘taking personal responsibility for engagement’ on behalf of those with diverse embodiments, or those who may not meet normative performance standards. In other words, participants uncovered valuable considerations for the PL development of individuals experiencing disability taking place within APA contexts, ones that potentially have a broader contextual applicability, and permit an expansion of PL as a meaningful idea for movers of all levels of ability.

Acknowledging that the meaning attached to the physical activity experiences of children is something organically produced through the relationships with others (i.e., interactive mesosystem influence between instructors, parents, children, and the larger community), instructors appreciated the value associated with a team-based approach to PL development. Collaborating with others and seeking input beyond their own ideas was essential to instructors’ understanding of PL, and ultimately, the creation of positive and customized learning climates that emphasized more than just skill acquisition, but were increasingly sensitive to the needs of each child (i.e., ‘pedagogical sensitivity’; [62]). Furthermore, being aware of these relationships was reflective of instructors’ beliefs to uphold a moral and ethical responsibility to ensuring that multiple voices were heard, physical activity opportunities reflected abilities, interests, and needs, and movement experiences were success-oriented toward the aim of sustained engagement [63]. Above all else, instructors were more about facilitating a positive experience for the child.

Challenging the dominant PL narrative dictated by the top-down, ableist culture of sport and broader society (exosystem and macrosystem influences), PL held meaning for instructors as a shared pedagogical responsibility [64]. Instructors recognized that PL development was not isolated to a particular context, nor was it their sole responsibility [65]. As such, they valued the importance of relationships across contexts in their understanding of PL (e.g., home and community environments; [24,66]), and the act of sharing of information to foster consistent and continuous personal growth (i.e., human flourishing; [67]) as children transition to other programs, both within their organization and elsewhere. In other words, instructors’ understandings of PL reflected an ongoing process of development broadly occurring throughout an individual’s lifespan [5,6], one that is facilitated beyond one’s own circle of influence. Such understandings not only expand upon the current connotations of PL and its overall contextual applicability, but they are indicative of an ethical commitment to optimizing meaningful experiences and maintaining a high standard of professional APA practice [26,63]. More so, they imply that perhaps, all programs attempting to foster PL development can benefit from the wisdom of APA instructors.

Despite challenging the dominant PL narrative and expanding the concept’s applicability, instructors recognized that their emphasis on the ‘physical’ (e.g., movement skill acquisition and assessment) reinforced an incomplete understanding of PL as a holistic concept [68], countering the underlying ableism present in instructor training programs [14,16]. The instructors’ recognition of the importance of diverse embodiment as central to their understanding of PL raises concerns about why we readily and rapidly promote a normative model of PL, one that directly counters our intention to include regardless of one’s level of embodiment. Such adoption only serves to create ethical tensions, and perhaps moral discomfort [69] on behalf of instructors, who fail to attach meaning of the intersection of diverse embodiments and motor competence to their understanding of PL and its holistic development. Furthermore, it calls into question the adequacy of education and training practices (exosystem influences) in preparing instructors to facilitate meaningful physical activity experiences for all children. Why are we educating based on a top-down model of development centred upon the notion of normativity and based on such a narrow contextual scope if there is such variation in humanity as a whole? Exploration and facilitating an awareness of PL beyond the ableist lenses of the culture of sport and broader society would not only provide greater recognition of this diversity, but a celebration of it leading to greater acceptance of varied forms of embodiment, and thus inclusivity. Moreover, it would avoid the development of an enlightened state of PL practice (‘enlightened PL’; [65]), whereby despite our best efforts, individuals with diverse embodied capabilities continue to be marginalized, their opportunities for PL development minimized, and societal inequities exacerbated. 

We acknowledge limitations to the study. Although we presented findings from only one instructional program, the instructors provided a homogeneous sample from which to gain in-depth understanding of PL for children who experience disability [41,55]. The first author’s prior knowledge of the participants may have aided rapport building, it could have influenced their forthcomingness in the interviews. To counter this, the participants were reminded that their information would not be discussed outside of the research team [58,59]. 

All in all, the meaning attached to PL, and the understandings that the instructors possessed, were embedded in the upper-level macrosystem and exosystem influences (e.g., the dominant, sport-based understandings of PL and the emphasis on normative societal standards) coupled with lower-level mesosystem and microsystem influences (e.g., relational exchanges with children, parents, and other professionals). These understandings reflected competing personal and environmental influences, illuminating the value instructors placed on PL, as well as their commitment to facilitating meaningful PL experiences for children experiencing disability. Using an ecological systems perspective, PL interpreted to be the acquisition of normative movement skill development having the potential to marginalize children with diverse embodiments. The important implications of the study lie with the need for flexible pedagogical strategies as instructors actively moved beyond understanding PL as movement competency. They incorporated motivational aspects of engagement by incorporating play in their instructional approaches, giving value to partial participation, abandoning rigid lesson plans to address children’s daily needs, and family involvement to support sustained engagement in activities of PL. Continued use of ecological frameworks is recommended to garner deepening insight into the nature of individual–environment interactions associated with understanding the meaning of PL in APA and inclusive contexts.

## Data Availability

Data unavailable due to privacy and ethical restrictions.

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
