# Peer review of "The Meaning of Physical Literacy for Instructors of Children Experiencing Disability, from an Ecological Systems Perspective"

_children, 2023, doi:10.3390/children10071185_

Round 1

Reviewer 1 Report

This was an excellent and well-written paper, strongly rooted in the literature. There is a high need to expand the physical literacy discourse outside of the dominant, sport/performance focused narrative that exists and this paper did an excellent job of doing so. I have a few minor notes for the authors to check, but otherwise have no revisions.

Lines 162-166: the inclusion criteria could be interpreted in a few ways. My understanding is the instructors had to have completed a minimum of 6 consecutive months of instruction within the past 24 months, but with the 6 months and 24 months in separate criteria it took me a few passes to garner that. Perhaps rephrase to increase readability.

Line 382: "fluences" appears to be a typo

Author Response

Thank you for your comments. We are happy that you enjoyed reading the manuscript. We have made some additional changes in light of your comments:

We have clarified the 6 months of instruction over the past 24 months in inclusion criteria (b). It now reads: "(b) held the role of instructor for six consecutive months or greater within the organization over the past 24 months.

The word "fluences" has been changed to 'influences'. Apologies for the typo.

Reviewer 2 Report

It is suggested to close with the research question or the purpose of the study.
The minutes or number of the protocol of acceptance of approval of the meeting must be indicated.
The URL of the following references must be indicated: Almond, L. & Whitehead, M. (2012). Berk, L.E. (2013). Campbell, FK (2017). Creswell, JW (2012). Eatough, V. & Smith, J.A. (2017). Lincoln, YS, Lynham, SA, & Guba, EG (2011). Maya, M.J. (2009). Patton, MQ (2015). Pushkarenko, K. (2022). Shinebourne, P. (2011). Smith, JA, Flowers, P. & Larkin, M. (2009). Sparkes, AC & Smith, B. (2014). Vickerman, P. & DePauw, K. (2010). Whitehead, M. (2010). Whitehead, M. (2019).  

Author Response

Thank you for your comments to the manuscript in question. In light of your comments, the following alterations have been made:

  • The first sentence in the final paragraph (while not explicitly re-stating the purpose of the study) does address the study's purpose/objective regarding meaning and understanding related to PL. It now reads: " All in all, the meaning attached to PL, and the understandings that the instructors possessed, were embedded in the upper-level macrosystem and exosystem influences (e.g., the dominant, sport-based understandings of PL and the emphasis on normative societal standards) coupled with lower-level mesosystem and microsystem influences (e.g., relational exchanges with children, parents, and other professionals)."
  • For anonymity purposes, we have decided not to include the ethics application number for the study (this is what we believe you are referring to in your comment about the "acceptance of approval".
  • If it is possible, we have added doi/url links to the references:
    • Almond, L. & Whitehead, M. (2012) (none available)
    • Berk, L.E. (2013). (URL inserted)
    • Campbell, FK (2017). (URL inserted)
    • Creswell, JW (2012). (URL inserted)
    • Eatough, V. & Smith, J.A. (2017). (URL inserted)
    • Lincoln, YS, Lynham, SA, & Guba, EG (2011). (URL inserted)
    • Mayan, M.J. (2009). (doi inserted)
    • Patton, MQ (2015). (URL inserted)
    • Pushkarenko, K. (2022). (URL inserted)
    • Shinebourne, P. (2011). (URL inserted)
    • Smith, JA, Flowers, P. & Larkin, M. (2009). (none available)
    • Sparkes, AC & Smith, B. (2014). (URL inserted)
    • Vickerman, P. & DePauw, K. (2010). (URL inserted)
    • Whitehead, M. (2010). (doi inserted)
    • Whitehead, M. (2019). (URL inserted)